# The Characterization of the *Phloem Protein 2* Gene Family Associated with Resistance to *Sclerotinia sclerotiorum* in *Brassica napus*

**DOI:** 10.3390/ijms23073934

**Published:** 2022-04-01

**Authors:** Rong Zuo, Meili Xie, Feng Gao, Wahid Sumbal, Xiaohui Cheng, Yueying Liu, Zetao Bai, Shengyi Liu

**Affiliations:** Key Laboratory of Biology and Genetics Improvement of Oil Crops, Oil Crops Research Institute of Chinese Academy of Agricultural Sciences, Ministry of Agriculture and Rural Affairs, Wuhan 430062, China; hu086zr@163.com (R.Z.); xiemeili0101@163.com (M.X.); gaofeng01@caas.cn (F.G.); sumbalwahid@gmail.com (W.S.); cxh5495@163.com (X.C.); lyy680608@126.com (Y.L.); liusy@oilcrops.cn (S.L.)

**Keywords:** phloem protein 2, *Brassica napus*, phylogenetic analysis, *Sclerotinia* disease resistance

## Abstract

In plants, phloem is not only a vital structure that is used for nutrient transportation, but it is also the location of a response that defends against various stresses, named phloem-based defense (PBD). Phloem proteins (PP2s) are among the predominant proteins in phloem, indicating their potential functional role in PBD. Sclerotinia disease (SD), which is caused by the necrotrophic fungal pathogen *S. sclerotiorum (Sclerotinia sclerotiorum)*, is a devastating disease that affects oil crops, especially *Brassica napus (B. napus),* mainly by blocking nutrition and water transportation through xylem and phloem. Presently, the role of *PP2s* in SD resistance is still largely estimated. Therefore, in this study, we identified 62 members of the *PP2* gene family in the *B. napus* genome with an uneven distribution across the 19 chromosomes. A phylogenetic analysis classified the *BnPP2s* into four clusters (I–IV), with cluster I containing the most members (28 genes) as a consequence of its frequent genome segmental duplication. A comparison of the gene structures and conserved motifs suggested that *BnPP2* genes were well conserved in clusters II to IV, but were variable in cluster I. Interestingly, the motifs in different clusters displayed unique features, such as motif 6 specifically existing in cluster III and motif 1 being excluded from cluster IV. These results indicated the possible functional specification of *BnPP2s.* A transcriptome data analysis showed that the genes in clusters II to IV exhibited dynamic expression alternation in tissues and the stimulation of *S. sclerotiorum*, suggesting that they could participate in SD resistance. A GWAS analysis of a rapeseed population comprising 324 accessions identified four *BnPP2* genes that were potentially responsible for SD resistance and a transgenic study that was conducted by transiently expressing *BnPP2-6* in tobacco (*Nicotiana tabacum*) leaves validated their positive role in regulating SD resistance in terms of reduced lesion size after inoculation with *S. sclerotiorum* hyphal plugs. This study provides useful information on *PP2* gene functions in *B. napus* and could aid elaborated functional studies on the *PP2* gene family.

## 1. Introduction

As the essential component in the vascular systems of plants, phloem provides an efficient pathway for the long-distance transportation of photosynthates and other signal molecules [1]. It constitutes a strategic location for mounting defenses against insects and pathogens [2]. The phloem structure is mainly composed of sieve elements, companion cells, parenchyma cells and phloem fibers [3]. Previous studies have shown that there are two predominant phloem proteins: phloem protein 1 and phloem protein 2 (PP1 and PP2) [4,5]. Both proteins play an important role in the establishment of phloem-based defense (PBD), which is induced by insect attacks [6] and other stresses [7], such as wounding and oxidative conditions [8]. Proteome studies in *B. napus* [9,10], *Cucurbita* [11,12], *Medicago truncatula* [13] and *solanum lycopersicum* [14] have suggested that phloem proteins may not only be involved in growth and development, but also in stress response and signal conduction. Mechanically, PP1 and PP2 are covalently cross-linked by a disulfide bond to form a high molecular weight polymer that closes the sieve pores, thereby creating a physical barrier to help to resist infection from pathogens when under stress [2,15].

PP2 (phloem protein 2) is a 49 KDa dimeric chitin-binding lectin that preferentially binds to N-acetylglucosamine [16]. In pumpkin, PP2 is an RNA-binding, defense-related, chitooligosaccharide-specific lectin that is highly expressed in the sieve elements and companion cells [15]. *PP2-like* genes have been identified in 17 angiosperms and gymnosperms, indicating that *PP2* genes are widely spread in plants [17], but the number of *PP2* genes varies greatly among species and ancient gene duplication may be the cause. In *Arabidopsis thaliana (A. thaliana)*, 30 *PP2* genes were identified with a cluster of 10 tandem repeats (*AtPP2-B1* to *AtPP2-B10*) in a 30-kb region [17]. In ramie (*Boehmeria nivea*), 15 *PP2* genes were characterized and all of them were sensitive to insects and fungal infection [18]. The specific *PP2* genes involved in PBD have been demonstrated in several reports. *AtPP2-A1* is a key gene in the PBD of *A. thaliana* that positively affects resistance to the green peach aphid [8], while *AtPP2-A5* affects plant defenses against mites through the modulation of hormonal signaling [19]. An analysis of *CsPP2B15* in citrus plants suggested that *CsPP2B15* may play an important role in the response to huanglongbing [20]. When the huanglongbing pathogen infects plants, the PP1–PP2 protein polymer complex blocks the sieve tube and causes a massive deposition of corpus callosum on the sieve plate, thus forming a protective layer in the phloem structure to prevent the further invasion of the growing pathogen [21]. Several reports have found that *PP2* genes are also involved in the defense against bacterial diseases in plants. *AN3*(*AtPP2-A2*), *AN4*(*AtPP2-A9*) and *AN5*(*AtPP2-A1*) all participate in plant stress responses, especially overexpressed *AN4* and *AN5*, which significantly improve resistance to *pseudomonas syringae* (*P. syringae*) compared to wild plants [22], and *AtPP2-B10* also showed the same pattern of resistance [23,24]. *PP2* genes also play important roles in abiotic stress; for example, *AtPP2-B1(AFA1)* is a novel F-box protein that negatively regulates drought stress and ABA signaling [25,26] and *AtPP2-B11* acts as a positive regulator in response to salt stress in *A. thaliana* [27]. Under salt stress, *AtPP2-B11* (as an SCF E3 ligase) upregulates the expression of annexins, represses ROS accumulation and affects Na ^+^ homeostasis to improve resistance to salt stress. In addition, *AtPP2-B11* specifically degrades SnRK2.3 to attenuate ABA signaling and the abiotic stress response in *A. thaliana* [28].

The Sclerotinia disease (SD) caused by *S. sclerotiorum* is a devastating disease for nearly all dicotyledons and some monocotyledon plants, especially soybean, oilseed rape, sunflower and many other crops. *B. napus* is one of the most important oil crops globally [29,30]. However, SD is the most severe disease for *B. napus* and results in a dramatic decrease in its quality and quantity. Previous studies have shown that SD resistance in plants is a quantitative trait that is controlled by polygenes with additive and partially dominant effects [31]. Therefore, it is necessary to explore genes that are potentially responsible for SD resistance from various perspectives. Considering the key role of PP2 in PBD, we postulate that *PP2s* may also function in SD resistance. However, studies of the role of *PP2s* in SD resistance have not yet been conducted. Therefore, in this study, we carried out a genome-wide investigation to identify the *PP2* genes in *B. napus* (*BnPP2*) and validated their function in SD, based on tissue expression, fungi induction expression and disease resistance examination through the transient expression of target genes in tobacco (*Nicotiana tabacum*). This study provides insights into the role of *PP2* in SD resistance and lays the foundation for further studies on *PP2* functions.

## 2. Results

### 2.1. Identification of BnPP2 Gene Family in B. napus

We identified 62 *PP2* genes with an intact PP2 domain in *B. napus* by using PF14299 as a query. The detailed information for each *BnPP2* gene is described in Table 1, including gene ID, chromosome, amino acids (AA) length, isoelectric point (pI), molecular weight (MW) and exon number. Among all *PP2* genes, 30 genes were located in the A sub-genome, while 31 were in the C sub-genome and one gene (*BnaUnng03010D*) was uncertain (Table 1). The protein length of the BnPP2s ranged from 145 amino acids (*BnaA08g15170D*) to 432 amino acids (*BnaC02g47880D*), with an average length of 267 amino acids. Overall, 45 (72%) *BnPP2s* consisted of 3 introns, the predicted theoretical pI ranged from 4.78 to 9.26 and the MW values varied from 17.22 to 49.23 kDa. The distribution of *BnPP2s* across the chromosomes was uneven. The chromosomes A02, A06, C02 and C07 had the most *BnPP2* genes, with eight, seven, five and eight *BnPP2* genes, respectively. While A04, A07, C04, C07 and C09 only contained one *BnPP2* gene and no *BnPP2s* were located on A05, A10 or C10. The gene duplication events of the *BnPP2* genes were detected based on BLAST and MCScanX analyses on the genome of *B. napus* (Appendix A, Appendix A). There was a cluster of seven tandem repeats (*BnaA02g26270D* to *BnaA02g26350D*) across a 55-kb region on A02. Local duplication events were also identified on the C02, A06 and C07 chromosomes. In total, 56.45% of the *BnPP2* genes (35 *BnPP2s*) originated from whole-genome duplication (WGD) or segmental duplications and 13 *BnPP2s* resulted from dispersed duplications. Additionally, nine tandem and seven proximal gene duplication types were found. Collectively, we identified 62 *BnPP2* genes in *B. napus* and their detailed information, including gene characteristics, chromosomal distribution and duplication type, was further elucidated.

### 2.2. The Phylogenetic Analysis of BnPP2 Genes

To further classify the *BnPP2s*, we constructed a phylogenetic tree using the PP2 proteins of *B. napus* (62), *A. thaliana* (30) and *O. sativa* (*Oryza sativa*) (39) (Figure 1). Based on clade support values and the topology of the phylogenetic tree, four subfamilies were categorized: I to IV. The gene number varied significantly among the four groups, with the highest number of genes (66) in cluster I and the fewest (9) in cluster II, which only contained six *BnPP2* genes and three *AtPP2* genes. In cluster I, the *BnPP2s* were close to *AtPP2s*, while almost all *OsPP2* genes were subdivided into two subgroups, indicating that those *OsPP2* genes evolved independently after the species split of *B. napus*, *A. thaliana* and *O. sativa*. In cluster I, *BnaA07g35800D* and *BnaC06g40720D* were the closest homolog of *AtPP2-B11* [27], five members (*BnaA02g26350D, BnaA06g34180D, BnaC07g21700D*, *BnaC07g21710D* and *BnaUnng03010D*) shared a high homology with *AtPP2-B10* [23,24] and eight genes (*BnaC02g34420D*, *BnaA02g26270D*, *BnaC07g21670D*, *BnaA06g34210D*, *BnaC07g21680D*, *BnaA06g34200D*, *BnaC07g21650D* and *BnaA06g34240D*) were clustered with *AtPP2-B1* [25], demonstrating that the *BnPP2* genes could have a function in stress-related processes [25,26,27,28]. In cluster III, 15 *BnPP2* genes were grouped with 5 *AtPP2* genes and 11 *OsPP2* genes and then 13 *BnPP2* genes in cluster IV were phylogenetically close to 10 *AtPP2* and 2 *OsPP2* genes. *BnaC01g41260D* and *BnaA01g10260D* were closely related to *AtPP2-2* [32] and both *BnaCnng21140D* and *BnaA09g25350D* shared a high sequence similarity to *AtPP2-9* [22], implying that they could have a similar function in insect and pathogen resistance.

To explore the evolutionary characteristics of *BnPP2* genes, we further compared the gene structure and conserved motifs between the four clades. The structures of most genes in clusters II–IV were well conserved (Figure 2c), which included three exons, except one gene in cluster III (*BnaC08g24190D*) and one gene in cluster IV (*BnaC09g52030D*). However, in cluster I, the gene structures varied remarkably from two to six, indicating the dynamic changes in gene structures within this clade. Then, we scanned the conserved motifs in all BnPP2s proteins and arranged them according to gene phylogenetic category (Figure 2a,b). The motifs 2 to 7, corresponding to the PP2 domain, were highly conserved among all BnPP2s (Figure 2b, Appendix A). However, similar to the gene structure, the motifs in cluster I varied significantly in terms of motif number. Additionally, most genes in this cluster contained motif 1 (F-box), except five genes (*BnaA08G15170D*, *BnaUnng03010D*, *BnaA02g26340D*, *BnaC07g21710D* and *BnaC02g47870D*). In clusters II–IV, the motifs were well conserved. Intriguingly, three genes in cluster II specifically contained motif 8 (*BnaC01g27190D*, *BnaAnng39370D* and *BnaC05g06650D*), which was an unknown domain. Genes in cluster III uniquely included motif 6, including 56–87 amino acid residue of the PP2 domain (Appendix A). Furthermore, all *BnPP2* genes in cluster IV excluded motif 1. These specific features implied the functional diversity of *BnPP2* genes.

### 2.3. Expression Analysis of BnPP2 in the Different Tissues and under the S. sclerotiorum Induction Condition

Gene expression patterns can reflect their potential function. Therefore, we then explored the expression profiles of the *BnPP2* genes using the available published transcriptome data [29,33]. The FPKM values for each *BnPP2* gene from five tissue samples (root, stem, leaf, bud and silique) were shown in Appendix A and the expression profiles were clustered across the five tissue samples (Figure 3a). Overall, 53 genes (53 out of 62) were expressed in at least one tissue, including 14 genes being highly expressed (FPKM > 50), 36 genes being medially expressed (FPKM > 1) and 3 genes being lowly expressed (FPKM < 1). When the expression levels were compared across the tissues, it was observed that the majority of *BnPP2* genes were substantially expressed in the roots, leaves, stems and buds, but were scarcely expressed in silique (Appendix A). Most genes in cluster I showed low expression, which was consistent with the frequent duplication and expansion. Among the genes in clusters II–IV, most *BnPP2* genes were highly expressed in the root (10 out of 34) and leaf (5 out of 34) (FPKM > 50), indicating their role in the plant’s response to environmental stimuli. Interestingly, five genes (*BnaC03g63410D*, *BnaA08g09350D*, *BnaC01g41230D*, *BnaA01g34970D* and *BnaC05g07510D*) in cluster IV exhibited extremely high expression in all tissues, implying their essential function in the development of *B. napus*. Further, we checked the expression differentiation of *BnPP2* under the induction of *S. sclerotiorum* using the published transcriptome data that were collected from the leaves of two cultivars [29], Zhouyou821 (ZY821) (resistant to SD) and Westar (susceptible to SD), at the time points of 0 and 24 h after inoculation with *S. sclerotiorum*. Consistent with the low expression of *BnPP2* in cluster I, they also showed minor transcript fluctuations after the induction of *S. sclerotiorum* (Appendix A). However, the majority of *BnPP2* genes (25 out of 34) in clusters II–IV exhibited remarkable expression changes in response to *S. sclerotiorum*, including 11 genes showing increased expression and 14 genes showing decreased expression 24 h post-inoculation (hpi) in at least one cultivar (Figure 3b, Appendix A). In cluster II, the expression of *BnPP2* was induced by *S.sclerotiorum*, except for two unexpressed genes (*BnaCnng75630D* and *BnaA08g26600D*). There were four genes (*BnaC03g63410D*, *BnaA08g09350D*, *BnaC01g41230D* and *BnaA01g34970D*) with high expression in both two cultivars (FPKM > 100) in cluster IV and their expression decreased at 24 hpi, especially in ZY821. The response pattern of *BnPP2* genes in cluster III was complicated, with the expression level of six genes being upregulated and nine genes being downregulated at 24 hpi in at least one cultivar. Interestingly, in ZY821, which is highly resistant to SD, most *BnPP2* genes showed more dramatic expression level changes compared to Westar, suggesting their possible role in the SD resistance of *B. napus*. For example, the expression of *BnaAnng39370D* and *BnaC05g06650D* in ZY821 was significantly more upregulated in ZY821 than in the Westar cultivar (Figure 3b, Appendix A).

Furthermore, we selected 12 candidate *BnPP2* genes, including 8 genes (*BnaC03g63410D*, *BnaA01g34970D*, *BnaA08g09350D*, *BnaC01g41230D*, *BnaC04g21970D*, *BnaA04g00910D*, *BnaA03g16790D* and *BnaA06g05940D*, named *BnPP2-1* to *BnPP2-8*) that displayed decreased expression and 4 genes (*BnaA09g10770D*, *BnaC07g04100D*, *BnaA01g35760D* and *BnaC05g06650D*, named *BnPP2-9* to *BnPP2-12*) that presented increased expression after fungi induction, to conduct experimental validation by quantitative real-time PCR (qRT-PCR). We collected leaves after inoculation with *S. sclerotiorum* to explore the dynamic gene expression changes at four time points (12 h, 24 h, 36 h and 48 h). The results showed that the expression levels of four genes, *BnPP2-9* to *BnPP2-12*, were upregulated at all of the time points after *S. sclerotiorum* inoculation, which was consistent with the transcriptome data (Figure 4). However, the expression changes of the other candidate genes were complex. After inoculation, the expression of *BnPP2-1* to *BnPP2-7* first increased and reached a peak at 24 hpi, then decreased at 36 and 48 hpi, while *BnPP2-8* reached its highest expression level at 12 hpi, then remarkably decreased at 24 hpi (Figure 4). These results indicated that the regulation of BnPP2-1 to BnPP2-8 for SD resistance was complicated. Therefore, a further sophisticated study is needed in this area.

### 2.4. Cis-Element and Protein Interaction Analysis of BnPP2 Genes

For a preliminary exploration of the potential mechanism of *BnPP2s*, we further analyzed the cis-elements of promoters and possible interactive proteins based on the available public database [34] (https://www.string-db.org/, accessed on 10 January 2022). The cis-elements were scanned across the promoters (2 kb) of *BnPP2s.* The results showed that these promoters were enriched with: hormone responsive elements (Appendix A), such as ABRE (involved in the abscisic acid response), CGTCA motif and TGACG motif (involved in the methyl jasmonic acid responsive MeJA response), AuxRR core and TGA element (involved in the auxin response), ERE (involved in the ethylene response) and the GARE motif, P-box and TATC-box (involved in gibberellin response); stress responsive elements, such as LTR (involved in low temperature response), MBS (involved in drought inducibility), DRE core (involved in dehydration response), WUN motif and TC-rich repeats (involved in wound response and defense and stress responses); and light responsive elements, such as Box4, G-box and GT1 motif (Appendix A). In clusters II to IV, there was a large number of hormone-related cis-acting elements in the promoters of *BnPP2* genes: the ABRE elements were found in maximum copies and detected in 26 of the 34 *BnPP2* gene promoters; MeJA-associated elements were found in 23 of the 34 *BnPP2s* promoters; and ERE-related elements were identified in 22 of the 34 *BnPP2s* promoters (Figure 5a, Appendix A). Among clusters II to IV, the *BnPP2s* in cluster III contained the most ABRE elements (5) and JA-associated cis-acting elements (4) on average, whereas the *BnPP2s* of cluster IV had the most ERE elements (2) on average (Appendix A). The number of stress-related cis-acting elements were relatively fewer in number and among all of the stress elements, MBS elements were present in 19 of the 34 *BnPP2* promoters (Figure 5a, Appendix A). These results implied that hormone pathways could mediate the function of *BnPP2* genes.

To explore the possible mechanism of *BnPP2*, the interaction networks were predicted based on the well-known protein interactions in *A. thaliana*. Using the *A. thaliana* orthologous *BnPP2s* genes as the query, we identified 1617 interactive proteins in the *A. thaliana* database, which corresponded to 5983 proteins in *B. napus*. Most BnPP2 proteins interacted with each other (Appendix A). To further elucidate the functional category of BnPP2 interactive proteins, we performed gene ontology (GO) and a Kyoto Encyclopedia of Genes and Genomes (KEGG) enrichment analysis. Interestingly, the GO terms of iron transport, response to mechanical and biotic stimulus and the defense response by callose deposition were significantly enriched, suggesting their role in plant defenses (Figure 5b, Appendix A). The KEGG enrichment analysis showed these interactive proteins were mainly involved in fundamental metabolism and the synthesis of important secondary metabolites (Appendix A, Appendix A). The metabolites, such as linolenic acid and carotenoid, were significantly enriched: linolenic acid is the substrate of PUFA and LOX (the key enzyme of JA biosynthesis) and carotenoid leads to the biosynthesis and metabolic decomposition of ABA (https://www.kegg.jp/kegg/, accessed on 10 January 2022) (Appendix A).

### 2.5. Functional Analysis of BnPP2s by Genome-Wide Association Analysis (GWAS) and Transgenic Strategy

To screen the *BnPP2* genes that are potentially responsible for SD resistance, we examined the GWAS results that were analyzed in our lab previously [35,36]. In total, four *BnPP2* genes (*BnaA01g35760D(BnPP2-11)*, *BnaA01g34970D(BnPP2-2)*, *BnaA03g16790D(BnPP2-7)* and *BnaA04g00910D(BnPP2-6)*) were identified in the loci associated with SD resistance (Figure 6 and Appendix A), indicating that they could contribute to SD resistance. Then, we further validated the function of *BnPP2-6* (*BnaA04g00910D*) in SD resistance using the transgenic strategy. We transiently overexpressed the *BnPP2-6* in tobacco leaves using the *Agrobacterium*-mediated infiltration method. After 60 h of cultivation [37,38] at 22 °C with a 16-h light and 8-h dark photoperiod, we inoculated detached leaves with *S. sclerotiorum* hyphal plugs in a chamber with a humidity of >85%. Then, we examined the disease lesion sizes at 24, 36 and 48 hpi. The results showed that the overexpression of *BnPP2-6-ox* in tobacco exhibited significantly smaller disease lesion sizes than the control (empty vector) and that the difference was much more obvious as the time points after inoculation progressed (Figure 7). This result suggested that *BnPP2-6* could indeed positively regulate SD resistance.

## 3. Discussion

Phloem is nutrient-rich and represents a unique ecological niche for a variety of pathogens. Some of the most destructive pathogens that feed on or live in phloem cause tremendous economic losses worldwide [39]. Meanwhile, plants also evolved PBD systems to combat pathogens [6]. PP2 proteins are the most abundant proteins in phloem and they play an important role in pathogen–plant interaction [18]. Researchers have found that the prokaryotic expression of purified CsPP2-A1 has significant inhibitory effects against *Botrytis cinerea* and *Phytophthora* infestans [40]. Overexpressed *AtPP2-A1*, *AtPP2-A9* and *AtPP2-B10* improve plant resistance to *P. syringae* [22,24].The genetic complexity and paucity of resistant germplasm represent serious challenges in studying SD resistance in rapeseed [35]. However, many researchers have made substantial progress over recent years, especially in defense-related genes that are involved in SD resistance [31]. In recent years, an accumulating body of research has shown that some transcription factors and kinases are related to SD resistance signaling pathways, such as *BnWRKY33* [41,42], *BnWRKY70* [43], *BnMPK3* [44], *BnMPK4* [45], *BnMPK6* [46] and *AtGDSL1* [35]. Pathogenesis-related (PR) proteins, such as lipid transfer proteins [47], defensins [48] and thaumatin-like proteins [47], play positive roles in SD resistance, as well as some secondary metabolites synthesis, such as *BnUGT74B1* [49] and lignin [50]. Researchers have also demonstrated that increasing the lignin content in the stem of *B. napus* is an important strategy to control Sclerotinia [51]. In general, great progress has been made in improving SD resistance in rapeseed, but more work still needs to be further extended. As a potentially disease resistant gene family, the role of PP2s in SD resistance has not yet been reported. So, in this study, we performed a comprehensive analysis of *BnPP2s* at the genome level and analyzed their evolutionary characteristics and functional impact on SD resistance. Therefore, the current study could facilitate new insights into this gene family and predict its potential function in plant stress conditions, especially in SD resistance.

In this study, we identified 62 *BnPP2s* in *B. napus* by using the Darmor-bzh v4.1 genome sequence information. The *BnPP2s* genes were unevenly distributed across the 19 chromosomes. Chromosomes A02, A06, C02 and C07 had the most *BnPP2* genes, while no *BnPP2s* were located on A05, A10 or C10 (Table 1). Frequent genome segmental duplication may be the cause of this. Phylogenetic analysis classified *BnPP2s* into four clusters (I–IV) and cluster I contained the most members (28 genes). Four tandem repeat gene clusters on chrC02, chrA02, chrA06 and chrC07 were identified in cluster I (Appendix A, Appendix A), suggesting that gene duplication could cause the proliferation of the BnPP2 cluster I. *B. napus* is an allotetraploid species formed by the hybridization of *B. rapa* and *B. oleracea* 7500 years ago, during which many rounds of duplication events have occurred [52,53]. Whole-genome duplication (WGD) plays a positive role in vascular plant speciation and is an essential mechanism for species to adapt to extreme environments [54]. Segmental duplications and WGD are also important for the production of duplicated genes and the expansion of gene families. This expansion was observed in the *BnPP2* genes as well (Figure 1, Appendix A). The evolutionary fate of duplicated genes includes non-functionalization, neo-functionalization or sub-functionalization [55]. The gene structures and conserved motifs of *BnPP2s* genes varied significantly in cluster I, which was not as well-conserved as clusters II–IV. Additionally, most *BnPP2s* in cluster I had very low expression, which differed from clusters II–IV. Tissue data showed 13 *BnPP2s* (13/28) were lowly expressed in almost all tissues in cluster I (FPKM < 1), while three genes (*BnaA07g35800D*, *BnaC06g40720D* and *BnaA02g26300D*) showed high expression in the root and stem tissues and *BnaA07g35800D* was only expressed in the root (Appendix A). Furthermore, we found 20 *BnPP2s* (20/28) had low expression or even almost no expression (FPKM < 1) in both cultivars, based on the transcriptome data from two rapeseed cultivars that were stimulated by *S. sclerotiorum* (Appendix A). These results indicated that functional differentiation occurred among some duplicate genes in cluster I. The number of *PP2s* is different between species, such as *A. thaliana* (30), *O. sativa* (39) and *Boehmeria nivea* (15) [17,18]. Gene duplication could cause these differences in the number of PP2 family members. Our results showed that duplication was the evolutionary force behind the *BnPP2* gene family expansion.

According to the phylogenetic analysis, the *BnPP2s* were classified into four classes along with *AtPP2s* and *OsPP2s* (Figure 1), suggesting a close evolutionary relationship between the three plant species. Most *OsPP2s* were then subdivided into three subgroups, except for three *OsPP2s* that were close to *AtPP2s* (Figure 1). This result revealed that PP2s were conserved in dicotyledons and monocotyledons. The structures of most *BnPP2s* in clusters II–IV were well conserved and the motifs were conserved in each clade (Figure 2b,c). Previous studies have shown that PP2 proteins have four conserved motifs (A, B, C and D), out of which motif B has the greatest diversity [17]. In our study, most BnPP2s had complete PP2 protein motifs, as with the four motifs described in cucurbit and celery PP2 proteins [17]. Almost all BnPP2s shared the central A motif (55/62) and the carboxy-terminal D motif (61/62), but motif B had a large diversity (Figure 2b, Appendix A), indicating that *BnPP2s* underwent a huge variation during evolution. The length of the inter-domain region was variable and explained the heterogeneity in most of the BnPP2s molecular weights (Table 1). As well as size polymorphism, BnPP2s also showed variation in electric charge (pI) (Table 1). The expression patterns of *BnPP2s* in different subfamilies also varied greatly. *BnPP2s* in cluster II were highly expressed in root tissue and in response to *S. sclerotiorum* (except for two unexpressed genes), while the *BnPP2s* of cluster III were highly expressed in root and bud tissues and some *BnPP2s* were upregulated and some *BnPP2s* were downregulated at 24 hpi in at least one cultivar after being infected by *S. sclerotiorum* (Figure 3a,b, Appendix A). This result suggested that the functions of *BnPP2s* have dramatically diverged, possibly due to the variation of motif B.

Based on the analysis of BnPP2 proteins within the domain search programs of the Conserved Domain Database (NCBI https://www.ncbi.nlm.nih.gov/cdd, accessed on 10 January 2022), we found that some BnPP2 proteins were multi-domain proteins. Two additional domains (TIR domain and F-box domain) were identified in the N-terminal regions of the BnPP2 proteins. We found that *BnaC09g52030D* and *BnaC08g49870D* presented a TIR domain that consisted of 170 residues in the N-terminus of the proteins in cluster IV. TIR domains that have been reported in plants are involved in the initial interactions with specific ligands that activate intracellular signaling cascades in response to pathogens [19]. Most of the *BnPP2s* in clusters I–III contained an F-box domain in the N-terminus, such as *BnaC02g47880D* with 432 amino acids, which had two PP2 domains and one F-box domain. As we all know, F-boxes are extremely widespread in plants and are widely involved in various physiological processes, including signal transduction, such as light signaling and hormone signaling, defense and stress responses and circadian rhythms [56]. In these processes, F-boxes are typically involved in targeting proteins in E3 ubiquitin-mediated degradation pathways [57]. Combined with the current transcriptomic data analysis, qRT-PCR results and GWAS analysis were previously completed in our lab and we selected *BnPP2-6* for the preliminary functional study (Figure 4 and Figure 6). We found that the transient expression of *BnPP2-6* in tobacco could improve the resistance of tobacco to *S. sclerotiorum* (Figure 7). The way in which *BnPP2-6* recognizes and regulates *S. sclerotiorum* still needs further exploration. PP2s are relatively conserved in plants, so we hypothesized that *BnPP2-6* could cross-link with PP1 through a disulfide bond to form a high molecular weight polymer that blocks the sieve pores, increases callose deposition and forms a physical barrier to slow down the infection of hyphae [20,39]. Meanwhile, the promoter region of *BnPP2-6* included a large number of ABRE and JA cis-acting elements (Appendix A). A protein–protein interaction analysis of the BnPP2s revealed that they could be involved in various biological processes through interactions with other proteins (Appendix A) and specifically, the N-terminus of BnPP2-6 contained an F-box domain that could make this protein a signaling molecule (as with the F-box gene) that could transmit signals in plant hormone-related signaling pathways when pathogens invade [56]. So far, little is known about the molecular regulatory mechanism of PP2 proteins. This study of the *BnPP2* gene family could be helpful in expanding the genetic resources of *S. sclerotiorum* resistance in *B. napus* and the mechanism of resistance of the *BnPP2* gene to *S. sclerotiorum* can be intensively studied in the future.

## 4. Materials and Methods

### 4.1. Identification and Phylogenetic Analysis of BnPP2 Gene Family in B. napus

The current genome sequence and annotation information for the rapeseed cultivar “Darmor-bzh v4.1” was obtained from the Brassicaceae Database [52] (BRAD, http://brassicadb.cn/, accessed on 5 January 2022). The sequence information of all *BnPP2s*, such as ID, CDS, proteins, chromosomal location and gff3 annotation, was collected from the genome data files of Darmor-bzh. For the HMM analysis, PF14299 was used as a query in the Pfam database (http://pfam.xfam.org, accessed on 5 January 2022). HMMER3.0 (http://www.hmmer.org/, accessed on 5 January 2022) was used to search for *PP2s* in the entire protein database of *B. napus* (the e-value was set to 1 × 10^−5^). The protein sequences of *PP2s* in *A.thaliana* were downloaded from the TAIR database (https://www.arabidopsis.org/, accessed on 5 January 2022) and those of the *PP2* genes in *O. sativa* were downloaded from the Pfam database. Then, the SMART databases [58] (http://smart.embl.de/, accessed on 5 January 2022) and the Conserved Domain Database (NCBI https://www.ncbi.nlm.nih.gov/cdd, accessed on 5 January 2022) were used for the verification of the PP2 domain in the identified PP2s. The redundant PP2s were excluded manually. The peptide length, molecular weight and isoelectric point of each BnPP2 protein were calculated using the online ExPasy program (http://www.expasy.org/, accessed on 5 January 2022).

The physical locations of the *BnPP2* genes on the chromosomes were obtained from the annotation of the *B. napus* genome. To identify the gene duplication events, BLASTP was used with the e-value of 1 × 10^−10^ to align the sequence and MCScanX [59] was used to detect the duplication patterns, including segmental and tandem duplication. The chromosomal locations and duplication events were visualized using the TBtools software [60].

### 4.2. The Phylogenetic Analysis of BnPP2 Genes

To gain insights into the evolutionary relationships between PP2 family members, multiple sequence alignments of the PP2 amino acids of *A. thaliana*, *O. sativa* and *B. napus* were performed using the ClustalW2 [61]. The phylogenetic tree was generated with the MEGA7 [62] program using the Neighbor-Joining (NJ) method with 1000 bootstrap replications. The tree was visualized using iTOL v6.5.2 (https://itol.embl.de/, accessed on 10 March 2022). The *BnPP2* genes were further categorized into different subgroups according to the homology of *PP2* genes in *A. thaliana* and *O. sativa*. Multiple Expectation Maximization for Motif Elicitation (MEME 5.4.1) [63] was used to analyze the conserved motifs in the BnPP2 proteins. For this objective, the following parameters were calibrated: a maximum of eight motifs with an optimal width of 6–50 amino acids. The remaining parameters were set to their default values. The identified motifs were annotated using the Pfam database (http://pfam.xfam.org/search, accessed on 10 January 2022). The TBtools software was used to visualize the gene and motif structures.

### 4.3. Expression Analysis of BnPP2 in Spatial-Temporal and S. sclerotiorum Induction Conditions

Transcriptome data from five tissues [33] (root, stem, leaf, bud and silique) of Zhongshuang 11 and two cultivars (susceptible *B. napus* vs. Westar and tolerant *B. napus* vs. Zhongyou821) under the induction of *S. sclerotiorum* fungi [29] were used in this study. The expression levels of the *BnPP2* genes were calculated with Stringtie [64]. Finally, the FPKM values were converted into log2 fold and the heat maps of all data were displayed by TBtools software.

The seeds of ZY821 were germinated and grown in a growth room at 22 °C with a 16-h light and 8-h dark photoperiod. When the rapeseed grew to the four or five leaf stage, they were prepared for inoculation with the hypha of *S. sclerotiorum*. First, the fungal strains preserved at 4 °C were sub-cultured onto a potato dextrose agar medium. Then, the new marginal hyphae were excised using a 7-mm puncher and were carefully upended onto the adaxial surface of healthy leaves. The inoculated plants were placed in a humidification chamber to keep the humidity above 85%. Each rapeseed plant was inoculated on three leaves and samples were taken every 12 h and immediately stored in liquid nitrogen. For each biological replicate, lesions were pooled from a minimum of three different plants and ground into a powder in liquid nitrogen. The total RNA was isolated using the Invitrogen TRIZOL Reagent (https://www.thermofisher.com, accessed on 10 January 2022). First-strand complementary DNA (cDNA) was synthesized using a TaKaRa reverse transcription kit (https://www.takarabiomed.com.cn/, accessed on 10 January 2022). To verify the response pattern of candidate *BnPP2* genes to *Sclerotiorum*, qRT-PCR primers of these candidate genes were designed (Appendix A). The qRT-PCR was carried out using Bio Supermix (http://www.bio-rad.com/, accessed on 10 January 2022) following the manufacturer’s instructions and reaction steps were performed as per the following program: 95 °C for 3 min; 40 cycles of 95 °C for 15 s; 56 °C for 15 s followed by 65 °C for 5 s and 95 °C for 5s. The *B. napus β*-actin gene (*AF111812*) was used as a reference standard. The relative expression was calculated using the 2^−∆∆*C*t^ method [65].

### 4.4. Cis-Element and Protein Interaction Analysis of BnPP2 Genes

The promoters of the *BnPP2* genes (2 kb upstream sequences from the initial codon) were extracted to identify the cis-acting regulatory elements in them using PlantCARE [34] (http://bioinformatics.psb.ugent.be/webtools/plantcare/html/, accessed on 10 January 2022). The protein–protein interactions (PPIs) of the PP2 proteins in *A. thaliana* were downloaded from STRING (https://www.string-db.org/, accessed on 10 January 2022), the functional association networks of the PP2 proteins in *B. napus* were predicted based on the homologs in *A. thaliana* and Cytoscape [66] was used to display the interactions. The genes that interacted with the BnPP2 proteins were taken out for gene ontology and KEGG enrichment analysis using the cluster Profiler in R [67].

### 4.5. Functional Analysis of BnPP2s by Genome-Wide Association Analysis (GWAS) and Transgenic Strategy

To screen the *BnPP2* genes that are potentially responsible for SD resistance, the GWAS results that were previously completed in our lab [35,36] were checked. The GWAS population, including 324 rapeseed accessions with different resistance levels, was collected from worldwide studies. The resequencing of these accessions was performed by the commercial Illumina HiSeq XTen service (BGI-Shenzhen, China). For the SD resistance identification, leaves from plants that were grown in a field in Wuhan in 2015 were excised at the of three to four leaf stage and incubated in a growth room after inoculation with *S. sclerotiorum*. The disease lesion sizes were examined at 12, 24, 36 and 48 hpi.

The CDS of the candidate genes was cloned from ZS11 into pCambia2300-GFP at BamHI and KpnI sites using the ClonExpress II One Step Cloning Kit (Vazyme Biotech Co., Ltd., Nanjing, China) (primers are shown in Appendix A). The recombinant plasmid and empty vector (negative control, pCambia2300-GFP) were transformed into *Agraobecterium tumefaciens* GV3101 competent cells (AngYu Biotech Co., Ltd., Shanghai, China). The tobacco seeds were germinated and grown in a growth room at 22 °C with a 16-h light and 8-h dark photoperiod. Briefly, the agrobacterium cells were harvested and adjusted to OD600 0.6 and then injected into four-week-old tobacco (*Nicotiana tabacum*) leaves using syringes [38]. Every tobacco plant was injected in three leaves and each plasmid was injected in 15 tobacco plants. The inoculated plants were incubated for 60 h at 22 °C in a growth room with a 16-h light and 8-h dark photoperiod [37,38]. Then, the leaves (at least 20 leaves for both expressed *BnPP2-6-ox* and the control) were excised and detached leaves were inoculated with *S. sclerotiorum* hyphal plugs in a chamber with a humidity of >85% and cultured in darkness. Each leaf was inoculated with one mycelium block, then the sizes of the disease spots were measured and photographed every 12 h. Then, the data were analyzed statistically and three replicates were performed for each experiment.

## 5. Conclusions

In this study, we performed a genome-wide analysis of the *BnPP2* gene family in *B. napus*, based on publicly available genome data. In total, 62 *BnPP2s* were identified and phylogenetically categorized into four clusters. Although the genes in cluster I were variable in terms of the aspects of gene number, structure, conserved motif and expressions, those in clusters II–IV were highly conserved. Considering the transcription alternation in response to fungi stimulation and the significant site in the GWAS for SD resistance, four *BnPP2* genes were screened. Further, we experimentally validated that the upregulated expression of *BnPP2-6* could inhibit the spread of SD. Our study verified the functional role of *BnPP2* in *S. sclerotiorum* disease resistance and also provided clues for the further mechanical exploration of the role of *BnPP2* in SD resistance. We foresee that these results will be of great value for the further functional characterization of the *BnPP2* gene family when considering genetic improvements in agronomic traits or stress tolerance in *B. napus*.

## Figures and Tables

**Figure 1 ijms-23-03934-f001:**
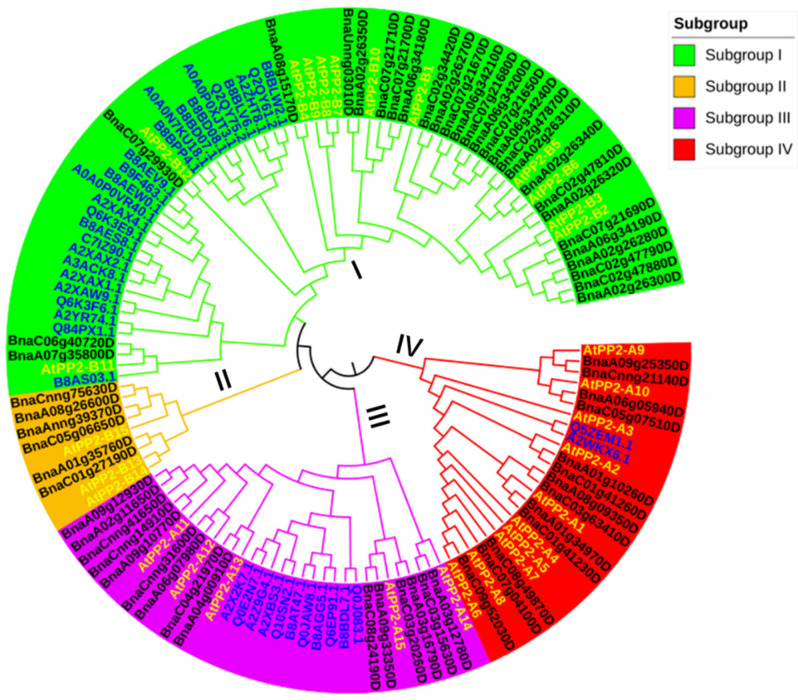
The phylogenetic analysis of PP2 proteins in *A. thaliana*, *O. sativa* and *B. napus.* All PP2 proteins were clustered into four subfamilies (I–IV) with differently colored branches (I, green; II, orange; III, purple; IV, red). The gene IDs for *B. napus* are black, the gene IDs for *A. thaliana* are bright yellow and those for *O. sativa* are blue.

**Figure 2 ijms-23-03934-f002:**
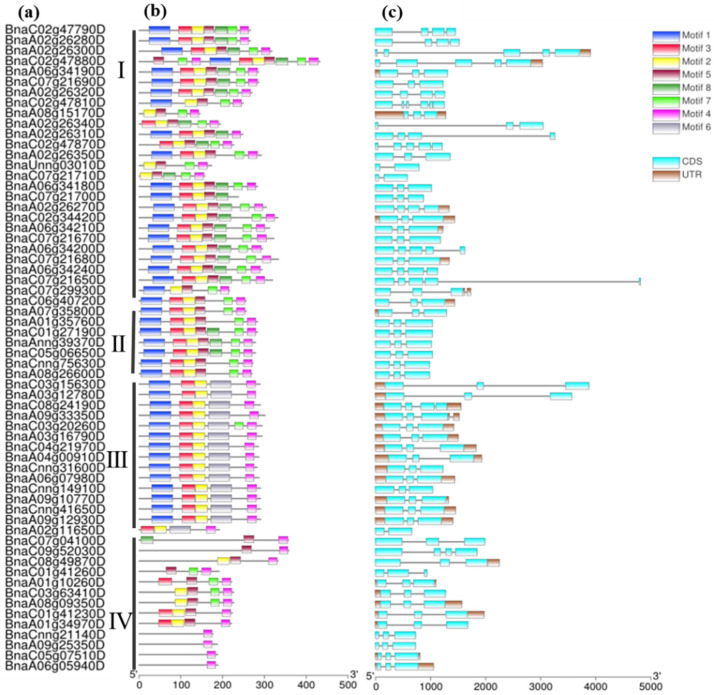
The gene structures and conserved motifs analysis of *BnPP2* genes: (**a**) gene ID; (**b**) conserved motif distribution in BnPP2 proteins. Numbers 1–8 are displayed in differently colored boxes. Motif 1 is the F-box domain and motifs 2–7 are part of the PP2 domain, located at 16–44 aa,1–15 aa, 126–146 aa, 45–64 aa, 56–78 aa and 98–116 aa, respectively. Motif 3 and motif 2 are labeled as motif A, motif 4 is motif D, motif 7 is motif C and motifs 5 and 6 is motif B. Appendix A contains more detailed information. (**c**) The gene structures of *BnPP2* genes. The dark brown boxes represent untranslated transcript regions (UTRs) and the blue boxes represent coding sequences (CDSs).

**Figure 3 ijms-23-03934-f003:**
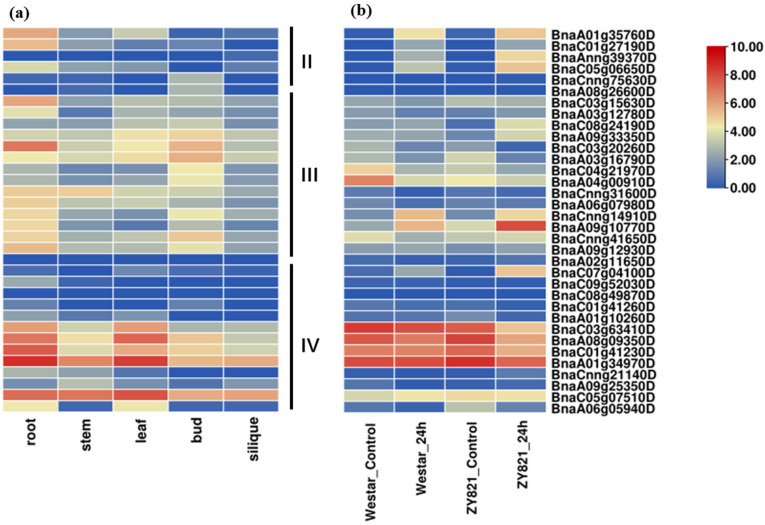
The expression patterns of the *BnPP2* genes in clusters II–IV: (**a**) the expression heatmap of *BnPP2* genes in various tissues from the root, stem, leaf, bud and silique; (**b**) the expression heatmap of *BnPP2* genes in Westar and ZY821 cultivars at 0 and 24 h after *S. sclerotiorum* inoculation. The expression data were processed with the log2 normalization of fragments per kilobase million (FPKM).

**Figure 4 ijms-23-03934-f004:**
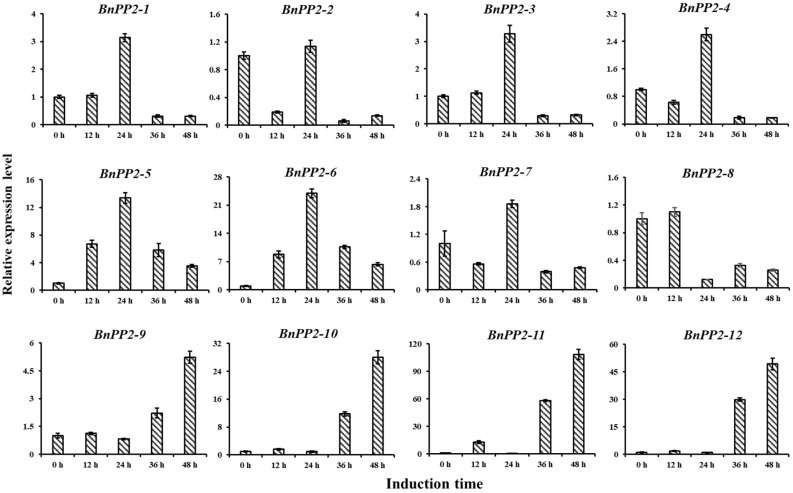
The expression validation of 12 candidate *BnPP2* genes in response to *S. sclerotiorum* by qRT-PCR. The time points 0 h, 12 h, 24 h, 36 h and 48 h represent hours after inoculation with *S. sclerotiorum*. The error bars show the standard error of three replicates.

**Figure 5 ijms-23-03934-f005:**
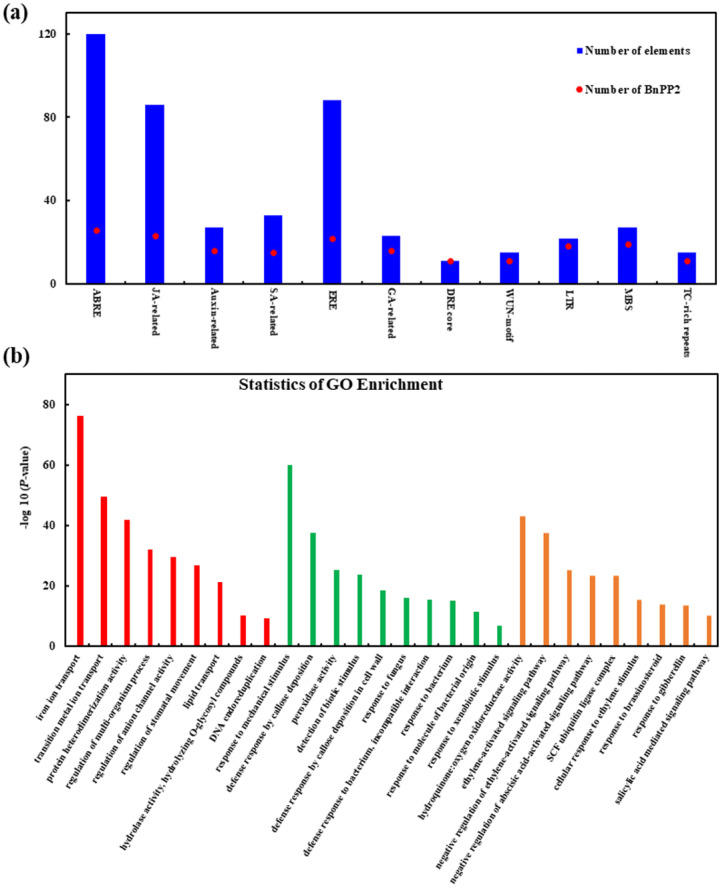
(**a**) The hormone-related and stress-related cis-acting regulatory elements in the promoters of the *BnPP2* genes in clusters II–IV. The bar graph indicates the total number of each cis-acting element found in the *BnPP2* promoters (royal blue box), as well as the number of *BnPP2* promoters that included a specific cis-regulatory element (red circle). (**b**) The gene ontology enrichment analysis of proteins that interacted with BnPP2 proteins. Appendix A contains more detailed information.

**Figure 6 ijms-23-03934-f006:**
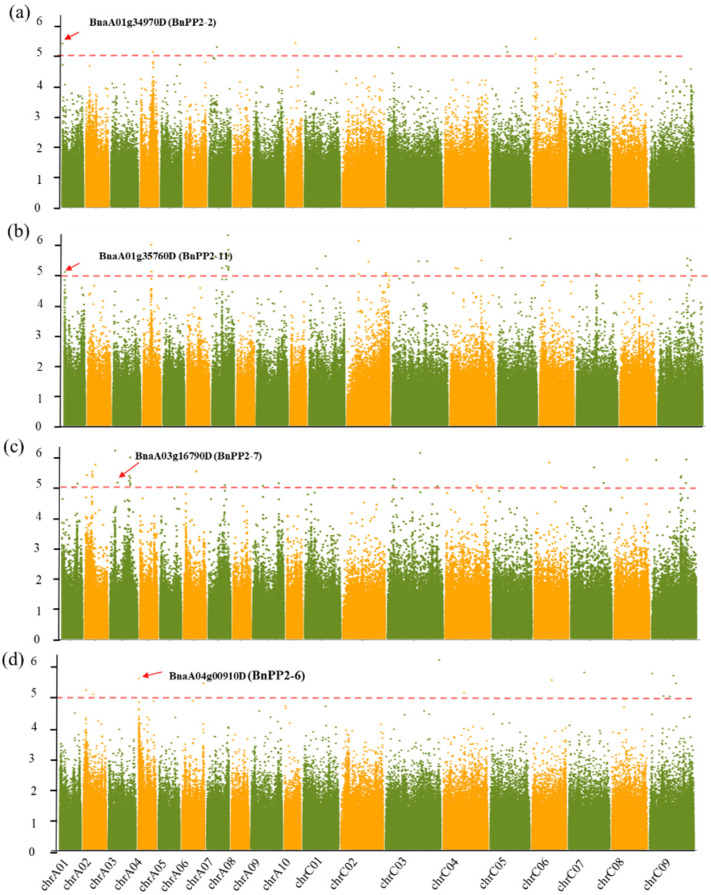
The genome-wide association analysis (GWAS) for SD resistance in a *B. napus* population comprising 324 accessions and Manhattan plots of SD resistance from association analyses: (**a**) Manhattan plots of the disease after 48–24 h; (**b**) Manhattan plots of the disease after 36–24 h; (**c**) Manhattan plots of the disease after 48 h; (**d**) Manhattan plots of the disease after 24 h. The red dashed line shows the GWAS threshold (1/SNP number).

**Figure 7 ijms-23-03934-f007:**
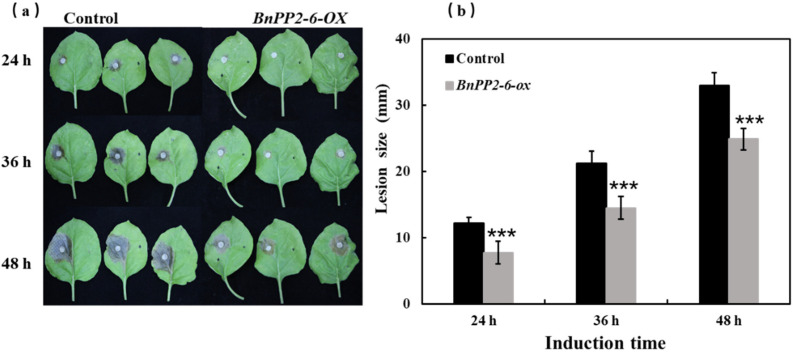
The functional validation of *BnPP2-6* for SD resistance using transient expression in tobacco: (**a**) the disease lesion sizes on leaves at 24, 36 and 48 h post inoculation with *S. sclerotiorum*, transiently injected through agrobacterium containing *BnPP2-6-ox* recombinant vectors and empty vectors (negative control) into tobacco leaves; (**b**) the disease lesion sizes were analyzed statistically by comparing the overexpressed *BnPP2-6* to the control. The data represent the means ± 2 SDs from three independent experiments, each containing 20 leaves. Significant differences in lesion sizes between *BnPP2-6- Ox* and the control are indicated (Student’s *t*-test) as follows: *** *p* < 0.001.

**Table 1 ijms-23-03934-t001:** A summary of the detailed characteristics of *BnPP2s*.

Gene ID	Subfamily	Chromosome	Amino Acids	MW (kDa) ^a^	pI ^b^	Exon Number	Duplication Type ^c^
BnaC02g47790D	BnPP2-I	C02 Random	266	29.66	5.14	4	Proximal
BnaA02g26280D		A02	266	29.66	5.38	4	WGD or Segmental
BnaA02g26300D		A02	317	35.85	6.99	5	Tandem
BnaC02g47880D		C02 Random	432	49.23	6.24	5	Tandem
BnaA06g34190D		A06	286	32.62	5.78	3	WGD or Segmental
BnaC07g21690D		C07	286	32.57	5.57	3	WGD or Segmental
BnaA02g26320D		A02	270	30.18	5.31	4	Proximal
BnaC02g47810D		C02 Random	249	28.74	8.13	6	Proximal
BnaA08g15170D		A08	145	17.22	5.02	3	Dispersed
BnaA02g26340D		A02	194	22.63	6.2	3	WGD or Segmental
BnaA02g26310D		A02	248	28.21	9.12	4	WGD or Segmental
BnaC02g47870D		C02 Random	226	26.17	7.53	5	Tandem
BnaA02g26350D		A02	292	33.58	6.04	3	Proximal
BnaUnng03010D		Unn Random	173	20.31	8.81	2	Dispersed
BnaC07g21710D		C07	157	18.43	9.26	2	Proximal
BnaA06g34180D		A06	283	32.04	4.78	3	WGD or Segmental
BnaC07g21700D		C07	237	26.78	5.3	3	Tandem
BnaA02g26270D		A02	304	34.11	5.32	4	WGD or Segmental
BnaC02g34420D		C02	332	37.34	5.46	3	WGD or Segmental
BnaA06g34210D		A06	312	35.24	5.13	3	Tandem
BnaC07g21670D		C07	322	36.22	5.18	3	Tandem
BnaA06g34200D		A06	295	33.05	5.11	5	WGD or Segmental
BnaC07g21680D		C07	333	37.55	5.7	3	Tandem
BnaA06g34240D		A06	294	32.24	5.08	4	Tandem
BnaC07g21650D		C07	319	35.18	5.02	5	WGD or Segmental
BnaC07g29930D		C07	216	24.3	4.79	3	Dispersed
BnaC06g40720D		C06	256	29.44	6.14	3	WGD or Segmental
BnaA07g35800D		A07	256	29.41	6.51	3	WGD or Segmental
BnaA01g35760D	BnPP2-II	A01 Random	283	31.93	6.37	3	Dispersed
BnaC01g27190D		C01	283	32.12	7.55	3	WGD or Segmental
BnaAnng39370D		Ann Random	278	31.68	5.83	3	Dispersed
BnaC05g06650D		C05	278	31.67	7.55	3	WGD or Segmental
BnaCnng75630D		Cnn Random	272	30.98	8.22	3	Dispersed
BnaA08g26600D		A08	270	30.53	8.22	3	WGD or Segmental
BnaC03g15630D	BnPP2-III	C03	289	33.12	8.63	3	WGD or Segmental
BnaA03g12780D		A03	278	31.79	8.82	3	WGD or Segmental
BnaC08g24190D		C08	290	32.75	5.31	4	WGD or Segmental
BnaA09g33350D		A09	301	33.92	5.86	3	WGD or Segmental
BnaC03g20260D		C03	294	33.39	8.01	3	WGD or Segmental
BnaA03g16790D		A03	294	33.41	7.56	3	WGD or Segmental
BnaC04g21970D		C04	285	32.68	9.14	3	WGD or Segmental
BnaA04g00910D		A04	286	32.75	9.24	3	WGD or Segmental
BnaCnng31600D		Cnn Random	282	32.06	7.06	3	WGD or Segmental
BnaA06g07980D		A06	287	32.62	8.01	3	WGD or Segmental
BnaCnng14910D		Cnn Random	290	32.94	8.95	3	WGD or Segmental
BnaA09g10770D		A09	290	32.86	8.95	3	WGD or Segmental
BnaCnng41650D		Cnn Random	290	33.08	8.89	3	Dispersed
BnaA09g12930D		A09	290	32.98	8.89	3	Dispersed
BnaA02g11650D		A02	192	22.09	8.65	2	Dispersed
BnaC07g04100D	BnPP2-IV	C07	357	41.66	7.02	3	Dispersed
BnaC09g52030D		C09 Random	358	41.22	6.24	4	Dispersed
BnaC08g49870D		C08 Random	332	37.78	5.88	3	WGD or Segmental
BnaC01g41260D		C01 Random	191	22.28	4.82	3	Proximal
BnaA01g10260D		A01	221	25.41	6	3	WGD or Segmental
BnaC03g63410D		C03	226	26.18	8.35	3	WGD or Segmental
BnaA08g09350D		A08	226	26.29	8.35	3	WGD or Segmental
BnaC01g41230D		C01 Random	223	25.77	8.34	3	WGD or Segmental
BnaA01g34970D		A01 Random	221	24.47	8.85	3	Dispersed
BnaCnng21140D		Cnn Random	176	19.76	8.98	3	WGD or Segmental
BnaA09g25350D		A09	187	20.95	7.85	3	WGD or Segmental
BnaC05g07510D		C05	188	21.35	9.24	3	WGD or Segmental
BnaA06g05940D		A06	188	21.37	9.26	3	WGD or Segmental

^a^ MW, molecular weight; ^b^ pI, isoelectric point; ^c^ duplication type; proximal, gene could arise from small-scale transposition or tandem duplication and the insertion of some other genes; WGD, whole genome duplication; dispersed, gene could arise from transposition, such as “replicative transposition”, “non-replicative transposition” or “conservative transposition”.

## Data Availability

The corresponding data have been shown in Appendix A.

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
