# Peer review of "The Characterization of the Phloem Protein 2 Gene Family Associated with Resistance to Sclerotinia sclerotiorum in Brassica napus"

_ijms, 2022, doi:10.3390/ijms23073934_

Round 1

Reviewer 1 Report

Zhou et al, have performed a comprehensive analysis of the Phloem protein 2 (PP2) gene family in the B. napus for their potential role in the resistance against the sclerotinia disease (SD) caused by necrotrophic fungal pathogen S. sclerotiorum. SD disease is one of the devastating diseases in oil crops. Authors have identified 62 BnPP2s in the B. napus by using the Darmor-bzh v4.1 genome sequence information and analyzed their evolutionary characteristics and functional impact on SD resistance. The manuscript is well written along with well planned methodology. The conclusions are supported by the results. Following are few points answering which might help in the overall manuscript quality:

  1. Authors have hypothesized that BnPP2-6 might crosslink with PP1 by a disulfide bond to form a high molecular weight polymer that blocks the sieve pores based on the transient expression of the BnPP2-6 in the tobacco leaves. The hypothesis is based on the evidence in the form of smaller disease lesion size. Does the tobacco PP1 has enough similarity with the BnPP1 protein, and will they be similar enough to form the PP1-PP2 dimers as expected by authors?
  2. Can authors give name to each of the identified genes in the beginning of the manuscript and use that name downstream rather than switching between genes original names and given names (qPCR data, selected gene for transient expression) at times? It would be lot easier to follow them.
  3. Gene names in figure 3 are not visible in white font color.
  4. It would be great for readers if the motif identified in the figure 2 be named in the figure or legend.
  5. In figure 2, exon-intron gene structure show some off the genes with or without UTR. Authors should make it clear whether the genes without UTRs are actually without UTR or its just based on gene prediction and they could be with UTRs.
  6. Figure 3 showing the expression data in the form of heatmap. Authors have not included the data from cluster I probably based on the low expression among those genes. Cluster I have lot of variability in gene structure and conserveness and have also gone through lot of duplications. It would be interesting to see whether the duplicated genes are functional or not?
  7. Please check the typo’s such as Line 53 (gene replication should be duplication), line 54 (A. thalina is repeated), etc.

Author Response

Thank you for the thorough review of the manuscript. We have revised the manuscript in detail according to your suggestions.

  1. Authors have hypothesized that BnPP2-6 might crosslink with PP1 by a disulfide bond to form a high molecular weight polymer that blocks the sieve pores based on the transient expression of the BnPP2-6 in the tobacco leaves. The hypothesis is based on the evidence in the form of smaller disease lesion size. Does the tobacco PP1 has enough similarity with the BnPP1 protein, and will they be similar enough to form the PP1-PP2 dimers as expected by authors?

This is a very good question and will be the focus of my future research. The phloem biology has not been extensively studied and the mechanism of phloem-based defense is unclear.  The molecular mechanism of BnPP2-6 in SD resistance need to further explore. PP2 is relatively conserved in plants. We compared the protein sequence of BnPP2-6 in the tobacco genome, and found that two proteins (A0A1S4DC56.1 and A0A1S4ARX2.1) were highly homologous to BnPP2-6, with 59% similarity, especially in the conserved domain. These results indicated that BnPP2-6 might form molecular sieves with endogenous PP1 protein of tobacco. We have proposed some hypotheses based on the literature and phenotype. These conjecture provide some ideas for our subsequent experimental design, which may be immature and need to be verified by more rigorous experiments designed in the future.

  1. Can authors give name to each of the identified genes in the beginning of the manuscript and use that name downstream rather than switching between genes original names and given names (qPCR data, selected gene for transient expression) at times? It would be lot easier to follow them.

Starting from 2.3, 12 candidate genes were named BnPP2 1-12, the given names are used in the following paragraphs. see lin2308-309 and Fig 6

  1. Gene names in figure 3 are not visible in white font color.

The white font in Figure 1 has been modified, see Fig 1.

  1. It would be great for readers if the motif identified in the figure 2 be named in the figure or legend.

The motifs in figure 2 are named in legend. Line 181-183

  1. In figure 2, exon-intron gene structure show some off the genes with or without UTR. Authors should make it clear whether the genes without UTRs are actually without UTR or its just based on gene prediction and they could be with UTRs.

The BnPP2 gene annotation was extracted from the GFF file of sequenced genome version of darmor-bzh (v4.1). Yes, there are some BnPP2 genes annotated as no UTR. However, further verification need to be emphasized, since the transcript of genes used for gene structure construction may not intact.

  1. Figure 3 showing the expression data in the form of heatmap. Authors have not included the data from cluster I probably based on the low expression among those genes. Cluster I have lot of variability in gene structure and conserveness and have also gone through lot of duplications. It would be interesting to see whether the duplicated genes are functional or not?

Yes, our study found that BnPP2 genes in cluster I is more variable compare to these in other clusters. Because of frequently duplication event occurred in genes of cluster I, it is reasonable to foreseen that these genes may underwent pseudogenization, sub- or neo- functionalization. However, the main focus of this study is prioritization the causal BnPP2 genes contributing to SD resistance, but few genes in cluster I showed expression alternation during the Sclerotinia fungi infection. We speculated that the genes of this subgroup might have no function in SD resistance. So, the gene expression data of the cluster I were not shown in the heat map and they were shown in detail in S2 and S3. Actually, I'm very curious about these duplicated genes, and it would be very interesting to study them, some of them could be playing important roles at a particular time in rape development, just waiting to be studied.

  1. Please check the typo’s such as Line 53 (gene replication should be duplication), line 54 (A. thalina is repeated), etc.

The appropriate changes have been made according to your suggestion (Line 53 Line 54). Thank you.

Reviewer 2 Report

The authors performra complete study on a family of proteins in a crop plants. Molecular studies on crop plants are not so frequent and are very useful both for plant biologists and agronomists. Also the phloem biology has not been extensively studied and is the path of entry for many plant pathogens, therefore I encourage publication.

Minor points.

Table 1: Rotulation. please put "amino" in one line and "acid" in other line.

Line 557: improvements in 

Author Response

Thank you for the thorough review of the manuscript. Thanks for your encouragement again, we will try our best to achieve a breakthrough in phloem resistance research

Table 1: Rotulation. please put "amino" in one line and "acid" in other line.

The appropriate changes have been made according to your suggestion (Table 1). Thank you.

Line 557: improvements in 

The appropriate changes have been made according to your suggestion (Line 557). Thank you.
